# Differentiable Unsupervised Feature Selection based on a Gated Laplacian

**Ofir Lindenbaum** *
Faculty of Engineering
Bar-Ilan University
Ramat Gan, Israel 5290002
`ofir.lindenbaum@biu.ac.il`

**Uri Shaham** *
Center for Outcome Research and Evaluation
Yale University
New Haven, CT 06510, USA
`uri.shaham@yale.edu`

**Erez Peterfreund**
Hebrew University

**Jonathan Svirsky**
Independent Researcher

**Nicolas Casey**
University of Pennsylvania

**Yuval Kluger**
Program in Applied Math
Department of Pathology
Yale University
New Haven, CT 06510, USA
`yuval.kluger@yale.edu`

## Abstract

Scientific observations may consist of a large number of variables (features). Selecting a subset of meaningful features is often crucial for identifying patterns hidden in the ambient space. In this paper, we present a method for unsupervised feature selection, and we demonstrate its advantage in clustering, a common unsupervised task. We propose a differentiable loss that combines a graph Laplacian-based score that favors low-frequency features with a gating mechanism for removing nuisance features. Our method improves upon the naive graph Laplacian score by replacing it with a gated variant computed on a subset of low-frequency features. We identify this subset by learning the parameters of continuously relaxed Bernoulli variables, which gate the entire feature space. We mathematically motivate the proposed approach and demonstrate that it is crucial to compute the graph Laplacian on the gated inputs rather than on the full feature space in the high noise regime. Using several real-world examples, we demonstrate the efficacy and advantage of the proposed approach over leading baselines.

## 1 Introduction

Unsupervised and self-supervised learning studies have been significantly growing interest in the neural network research community. This was prompted by the impressive empirical results that unsupervised learning methods produce in analyzing large amounts of unlabeled data, such as in natural language processing. Many scientific domains, such as biology and physics, have seen the growth of computational and storage resources, alongside advancements in the simultaneous measurement of numerous features, making the analysis of large, high-dimensional datasets a vital research need.

---

*Indicates equal contribution

35th Conference on Neural Information Processing Systems (NeurIPS 2021).

In such datasets, discarding irrelevant (e.g., noisy, high frequency, or information-poor) features may reveal clear underlying natural structures that are otherwise hidden in the high dimensional space. We refer to features which do not correspond to the "main" data structure as "nuisance features"[2]. While nuisance features are mildly harmful in the supervised regime, discarding such features is critical in the unsupervised regime and may determine the success of downstream analysis tasks (e.g., clustering or manifold learning). Some of the pitfalls caused by nuisance features could be mitigated using an appropriate unsupervised feature selection method.

Unsupervised feature selection methods mostly focus on one of two main tasks: either clustering or manifold learning. Among studies that tackle the former task, [33, 12, 1] use autoencoders to identify features that are sufficient for reconstructing the data. Other clustering-dedicated unsupervised feature selection methods assess the relevance of each feature based on different statistical or geometric measures. For example entropy, variance, divergence, and mutual information are used in [2, 31, 9, 3, 30, 35] to identify features that are informative for clustering the data. A popular tool for evaluating features is the graph Laplacian [26, 4]. The Laplacian Score (LS) [13] evaluates the importance of each feature by its ability to preserve the local structure. The features that most preserve the manifold structure (captured by the Laplacian) are retained. Several studies, such as [38, 27, 39], extend the LS based on different spectral properties of the Laplacian.

While these methods are widely used in the feature selection community, they rely on the success of the Laplacian in capturing the "main" structure of the data. We argue that the Laplacian often fails to identify the informative features when computed on the entire dataset (as is demonstrated in Section 3). This may happen in the presence of a large number of nuisance features: when the variability of the nuisance features masks the variability associated with the structured features. Scenarios resembling this are prevalent in bioinformatics for example, where a large number of biomarkers are measured to characterize developmental and chronological biology processes such as cell differentiation or cell cycle. These processes may depend merely on a few biomarkers. In such situations, it is desirable to have an unsupervised method that can filter nuisance features before the computation of the Laplacian. Related problems, in which data spanning low dimensional subspace is dressed by perturbations in a higher ambient space, were studied in [17, 5, 10, 28].

In this study, we propose a differentiable objective for unsupervised feature selection. Our proposed method utilizes stochastic input gates, trained to select features with high correlation with the leading eigenvectors of a graph Laplacian computed based on these features. This gating mechanism allows us to re-evaluate the Laplacian for different subsets of features and, thus, unmask the "main" data structure buried by the nuisance features. We demonstrate that the proposed approach can significantly improve cluster assignments compared with leading baselines using high-dimensional datasets from multiple domains (image, text, and biological observations).

## 2 Preliminaries

Consider a data matrix, $\boldsymbol{X} \in \mathbb{R}^{n \times d}$, with $d$ dimensional observations $\boldsymbol{x}_1, ..., \boldsymbol{x}_n$. We refer to the columns of $\boldsymbol{X}$ as features $\boldsymbol{f}_1, ..., \boldsymbol{f}_d$, where $\boldsymbol{f}_i \in \mathbb{R}^n$, and, assume that features are centered and normalized such that $\mathbf{1}^T \boldsymbol{f}_i = 0$ and $\|\boldsymbol{f}_i\|_2^2 = 1$. We assume that the data has an inherent structure, determined by a subset of the variables $\mathcal{S}^*$ and that other variables are nuisance features (i.e., noisy or information-poor). The structured part of the data is considered as either a path-connected manifold, a union of sub-manifolds, or a set of clusters. Our goal is to identify the subset of relevant features $\mathcal{S}^*$, which preserve the data structure and discard the remaining ones.

### 2.1 Graph Laplacian

Given $n$ data points, a kernel matrix $\boldsymbol{K}$ is an $n \times n$ matrix whose $\boldsymbol{K}_{i,j}$ entry represents the similarity between $\boldsymbol{x}_i$ and $\boldsymbol{x}_j$. A popular choice for $\boldsymbol{K}$ is the Gaussian kernel

$$\boldsymbol{K}_{i,j} = \exp\left(-\frac{\|\boldsymbol{x}_i - \boldsymbol{x}_j\|^2}{2\sigma_b^2}\right), \tag{1}$$

---

[2]In section 3, we consider scenarios where low dimensional datasets that form manifold or cluster structures are augmented with information-poor nuisance features that are independent of the original structure and are, themselves, of no particular underlying structure.

where $\sigma_b$ is a user-defined bandwidth (chosen, for example, based on the maximum value of the 1-nearest-neighbors of all points). The unnormalized graph Laplacian matrix is defined as $L_{\text{un}} = D - K$, where $D$ is a diagonal matrix, whose elements $D_{i,i} = \sum_{j=1}^{n} K_{i,j}$ correspond to the degrees of the points $i = 1, ..., n$. The diffusion graph Laplacian is defined as

$$L_{\text{diff}} = D^{-1}K, \tag{2}$$

and expresses the transition probabilities of a random walk to move between data points and induces a diffusion distance [8]. Graph Laplacian matrices are extremely useful in many unsupervised machine learning tasks. In particular, it is known that the eigenvectors corresponding to the small eigenvalues of the unnormalized Laplacian (or the large eigenvalues of the random walk Laplacian) are useful for embedding the data in lower dimensions (see, for example, [32]).

## 2.2 Laplacian Score

Following the success of the graph Laplacian [4], and [26], the authors in [13] have presented an unsupervised measure for feature selection, termed Laplacian Score (LS). The LS evaluates each feature based on its correlation with the leading eigenvectors of the graph Laplacian.

At the core of the LS method, the score of feature $f$ is determined by the quadratic form $f^T L_{\text{un}} f$, where $L_{\text{un}}$ is the unnormalized graph Laplacian. Since

$$f^T L_{\text{un}} f = \sum_{i=1}^{n} \lambda_i \langle u_i, f \rangle^2,$$

where $L_{\text{un}} = \sum_{i=1}^{n} \lambda_i u_i u_i^T$ is the eigen-decomposition of $L_{\text{un}}$, the score is smaller when $f$ has a larger projection on the subspace of the leading eigenvectors (corresponding to the smallest eigenvalues) of $L_{\text{un}}$. Such features can be thought of as "informative", as they respect the graph structure. Eigenvalues of the Laplacian can be interpreted as frequencies, and eigenvectors corresponding to larger eigenvalues of $L_{\text{un}}$ (or smaller eigenvalues of $L_{\text{diff}}$) oscillate faster. Based on the assumption that the interesting underlying structure of the data (e.g. clusters) depends on the slowly varying features in the data, [13] proposed to select the features with the smallest scores.

# 3 Demonstration of the Importance of Unsupervised Feature Selection in High Dimensional Data with Nuisance Features

By taking a diffusion perspective, we first demonstrate the importance of feature selection to unsupervised learning when the data contains nuisance features. Then, a simple two cluster structure hidden in the ambient space is used to analyze how Gaussian nuisance dimensions affect clustering capabilities. Here, we model the effect of nuisance variables by concatenating variables drawn from a normal Gaussian distribution with the structured part of the data. A concrete example is detailed in the following subsection.

## 3.1 A Diffusion Perspective

Consider the structured 2-dimensional dataset, known as two-moons, shown in the top-left panel of Fig. 1. We concatenate the structured data with $k$ "nuisance" dimensions, where each such dimension is composed of i.i.d unif$(0, 1)$ entries. This means that our new data consists of $2 + k$ variables. Note that the $k$ Gaussian variables do not carry any information about the two "moon-shaped" clusters in this dataset. As one may expect, when the number of nuisance dimensions is large, the amount of unstructured information (manifested by the nuisance dimensions) dominates the amount of structured information (manifested by the two leading dimensions). Consequently, attempts to recover the main structure of the data (say, using manifold learning or clustering) are likely to fail.

From a diffusion perspective, data is considered to be clusterable when the time it takes a random walk starting in one cluster to transition to a point outside the cluster is long. These *exit times* from different clusters are manifested by the leading eigenvalues of the Laplacian matrix: $L_{\text{diff}} = D^{-1}K$, for which the large eigenvalues (and their corresponding eigenvectors) are the ones that capture different aspects of the data's structure (see, for example, [24]). Each added nuisance dimension increases the distance between points which are nearest neighbors in the two dimensional structured subspace. In

addition, the noise creates spurious similarities between points, regardless of the cluster they belong to. Overall, this shortens the cluster *exit times*. This phenomenon is characterized by the second largest eigenvalue, $\lambda_2$ of $\boldsymbol{L}_{\text{diff}}$ (the largest eigenvalue $\lambda_1 = 1$ carries no information as it corresponds to the constant eigenvector $\boldsymbol{\psi}_1$), which decreases as the number of nuisance dimensions grows, see the top right panel of Fig. 1. The fact that $\lambda_2$ decreases implies that the diffusion distances [24] decrease as well which, in turn, means that the clusters become more connected. A similar view may be obtained by observing that the second smallest eigenvalue of the un-normalized graph Laplacian $\boldsymbol{L}_{\text{un}} = \boldsymbol{D} - \boldsymbol{K}$, also known as Fiedler number or algebraic connectivity, grows with the number of nuisance features. The fact that the graph becomes less clusterable as more nuisance dimensions are added is also manifested by the eigenvector, $\boldsymbol{\psi}_2$, corresponding to the second largest eigenvalue of $\boldsymbol{L}_{\text{diff}}$ (or the second smallest eigenvalue of $\boldsymbol{L}_{\text{un}}$), which becomes less representative of the cluster structure (bottom left panel of Fig. 1).

Altogether, this means that, in order for the data to be clusterable, the nuisance features ought to be removed. One may argue that principal component analysis can be used to retain the structure of the two-moons while attenuating the effect of the $k$ nuisance variables. However, as shown in the bottom right panel of Fig. 1, projecting the data onto the first two principal directions does not yield the desired result, since the directions of maximal variance do not capture the two-moons structure, unfortunately. In the next sections, we will describe our differentiable unsupervised feature selection approach, demonstrating that it does succeed to recognize the cluster patterns of the data in this case.

## 3.2 Analysis of Clustering with Nuisance Dimensions

In order to observe the effect of nuisance dimensions, this section considers a simple example with two clusters in 1-D space and a set of nuisnace variables which do not carry any information about these clusters. Specifically, consider a dataset that includes $2n$ datapoints in $\mathbb{R}$, where $n$ of which are at $0 \in \mathbb{R}$ and the remaining ones are at $r > 0$ (i.e., each cluster is concentrated at a specific point). Next, we concatenate $d$ nuisance dimensions to first coordinate, so samples lie in $\mathbb{R}^{d+1}$. The value for each datapoint in each nuisance dimension is sampled independently from $N(0, 0.5^2)$.

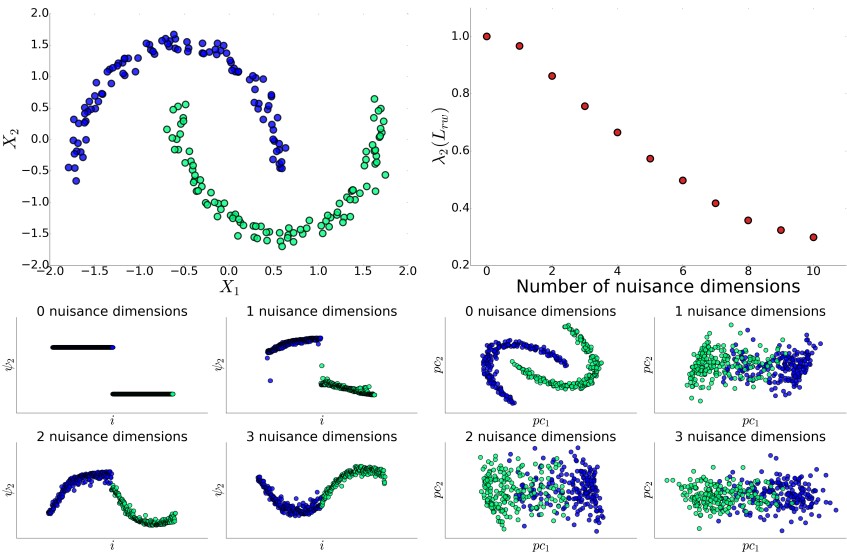

Figure 1: The Two-moons example. Top Left: the original 2-dimensional data. Top right: the second largest eigenvalue of the random walk matrix, $\boldsymbol{L}_{\text{diff}}$, decreases as the number, $k$, of nuisance dimensions grows. This suggests that the graph becomes more connected—and, hence, less clusterable—as the number of nuisance dimensions grows. Bottom left: the eigenvector, $\boldsymbol{\psi}_2$ (y-axis) (corresponding to the second largest eigenvalue), of $\boldsymbol{L}_{\text{diff}}$ becomes less representative of the true cluster structure as the number of nuisance variables ($k$) grows. The x-axis represents the sample index $i$. Bottom right: the leading two principal components of the $k + 2$ dimensional data (x and y axes) cannot help recover the true cluster structure when $k > 0$.

Suppose we construct the graph Laplacian by connecting each point to its nearest neighbors. We would now investigate the conditions under which the neighbors of each point belong to the correct cluster. Consider points $x, y$ belonging to the same cluster. Then $(x - y) = (0, u_1, \ldots, u_d)$ -where $u_i \overset{\text{iid}}{\sim} N(0,1)$, and, therefore, $\|x - y\|^2 \sim \chi_d^2$. Similarly, if $x, y$ belong to different clusters, then $\|x - y\|^2 \sim r^2 + \chi_d^2$, where $r$ is the distance between clusters in the first coordinate. Now, to find conditions for $n$ and $d$ under which the neighbors of each point are highly likely to belong to the same cluster, we can utilize the $\chi^2$ measure-concentration bounds [18].

**Lemma 3.1** ([18] P.1325). *Let $X \sim \chi_d^2$. Then*

1. $\mathbb{P}(X - d \geq 2\sqrt{d\gamma} + 2\gamma) \leq \exp(-\gamma)$.

2. $\mathbb{P}(d - X \geq 2\sqrt{d\gamma}) \leq \exp(-\gamma)$.

Given sufficiently small $\gamma > 0$ we can divide the segment $[d, d + r^2]$ to two disjoint segments of lengths $2\sqrt{d\gamma} + 2\gamma$ and $2\sqrt{d\gamma}$ (and solve for $d$ in order to have the total length $r^2$). This yields

$$\sqrt{d} = \frac{r^2 - 2\gamma}{4\sqrt{\gamma}}. \tag{3}$$

If all distances between points from the same cluster are at most $d + 2\sqrt{d\gamma} + 2\gamma$ and all distances between points from different clusters will be at least $d + r^2 - 2\sqrt{d\gamma}$, the nearest neighbors of each point will be from the same cluster. According to lemma 3.1, this happens with a probability of at least $(1 - \exp(-\gamma))^{2n^2 - n}$. For a small $\epsilon > 0$, denoting this probability as $1 - \epsilon$ and solving for $\gamma$, we obtain

$$\gamma \leq -\log(1 - \sqrt[(2n^2 - n)]{1 - \epsilon}). \tag{4}$$

Plugging (4) into (3), we obtain

$$d = O\left(\frac{r^4}{-\log(1 - \sqrt[(2n^2 - n)]{1 - \epsilon})}\right). \tag{5}$$

For fixed number of samples ($n$) and a small value of $\epsilon > 0$, equation (5) implies that the number of nuisance dimensions must be at most on the order of $r^4$ to make cluster mixture unlikely. In addition, for a fixed $r$ and $\epsilon$, increasing the number of data points brings the argument inside the log term arbitrarily close to zero, which implies that the Laplacian is sensitive to the number of nuisance dimensions in large datasets. We support these findings via experiments, as shown in Figure 2. This sensitivity to nuisance dimensions suggests that capturing the "main" structure of the data, requires filtering nuisance features prior to the construction of the Laplacian. In Section 4, we present our proposed approach, which simultaneously filters nuisance features and re-evaluates the smoothness of the remaining features.

## 4 Proposed Method

### 4.1 Rationale

Recall that the core component of the Laplacian score [13] is the quadratic term $\boldsymbol{f}^T \boldsymbol{L} \boldsymbol{f}$, which measures the inner product of the feature $\boldsymbol{f}$ with the eigenvectors of the Laplacian $\boldsymbol{L}$. For $\boldsymbol{L} = \boldsymbol{L}_{\text{diff}} = \boldsymbol{D}^{-1} \boldsymbol{K}$, a large Laplacian score implies that $\boldsymbol{f}$ has a large component in the subspace of eigenvectors corresponding to the largest eigenvalues of $\boldsymbol{L}$. Assuming that the structure of the data varies slowly, these leading eigenvectors (corresponding to large eigenvalues) manifest the main structures in the data; thus, a large score implies that a feature contributes to the structure of the data. However, as we demonstrated in the previous section, these leading eigenvectors become less representative of the true structure in the presence of nuisance features. In this regime, one could benefit from evaluations of the Laplacian score when it is computed using different subsets, $\mathcal{S}$, of features (i.e., of the form $\boldsymbol{f}^T \boldsymbol{L}_{X_{\mathcal{S}}} \boldsymbol{f}$, where $\boldsymbol{L}_{X_{\mathcal{S}}}$ is the random walk Laplacian computed based on a subset of features $\{\boldsymbol{f}_\ell\}_{\ell \in \mathcal{S}}$). When the Laplacian is computed on only the informative features (i.e., when $\boldsymbol{L}_{X_{\mathcal{S}}} = \boldsymbol{L}_{X_{\mathcal{S}^*}}$), such a gated Laplacian score would produce a high value for the informative features, $\mathcal{S}^*$.

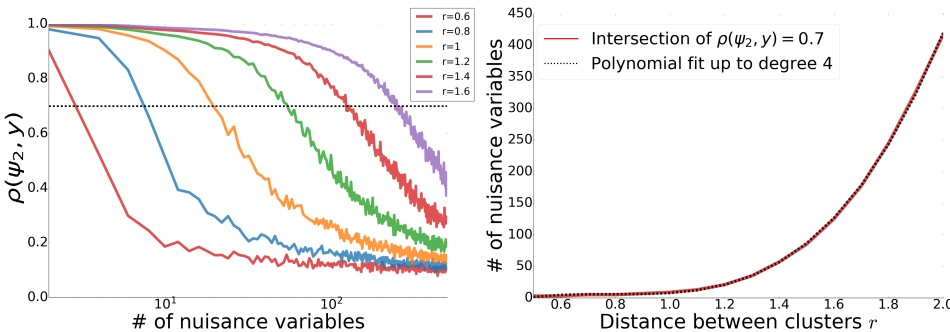

Figure 2: Two cluster datasets. We evaluate the influence of Gaussian nuisance variables on the Laplacian. We generate two clusters using $50$ samples each with distance $r$ apart in $1$-D. We use $d$ Gaussian nuisance variables and evaluate the leading non trivial eigenvector $\psi_2$ of the Laplacian. Left: correlation between the second eigenvector $\psi_1$ and the true cluster assignments $y$ for different values of $r$. As the number of nuisance variables grows, the eigenvector $\psi_2$ becomes meaningless. As the distance between cluster grows more nuisance variables are required to "break" the cluster structure captured by $\psi_2$. Right: by computing the intersection between the damped correlation curves and $0.7$ (shown in the left plot) for different values of $r$ we evaluate the relation between $r$ and number of nuisance variables $d$ required for breaking the cluster structure. This empirical result supports the analysis presented in 3.2 in which we show that $d = O\left(\dfrac{r^4}{-\log(1 - {}^{(2n^2-1)}\!\sqrt{1-\epsilon})}\right)$. For convenience we added a polynomial fit up to degree $4$ presented as the black line.

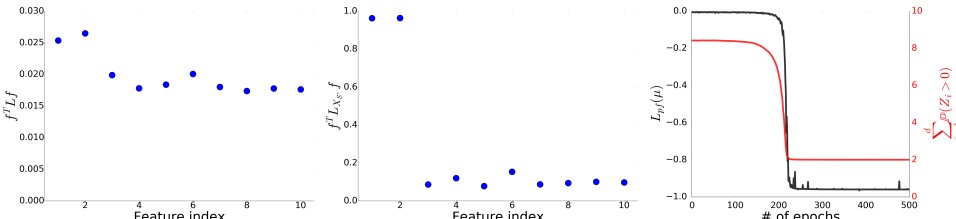

Figure 3: Illustrating the information captured by the Laplacian Score (using $L_{rw}$) on the noisy two-moons dataset (see Fig. 1). The first two features are informative, while the rest are nuisance variables. Our goal is to train the differentiable stochastic gates for identifying the two informative features. Left: Laplacian score $\boldsymbol{f}^T \boldsymbol{L} \boldsymbol{f}$ at initialization, based on all $10$ dimensions. The score for the informative features is slightly higher. Middle: Laplacian score, based on the gated Laplacian $\boldsymbol{f}^T \boldsymbol{L}_{X_{S*}} \boldsymbol{f}$ at the gates' convergence. The informative features attain a substantially higher score based on the gated Laplacian. Right: the parameter-free loss (black line) and the average number of active gates (red line) as a function of the number of epochs.

Searching over all the different feature subsets is infeasible even for a moderate number of features. Fortunately, we can use continuous stochastic "gating" functions to explore the space of feature subsets. Specifically, we propose to apply differential stochastic gates to the input features, computing the Laplacian score after multiplying the input features with the gates. Taking advantage of the fact that informative features are expected to have higher scores than nuisance ones, we penalize open gates to minimize the number of features and retain the most relevant (smooth) features. By applying gradient decent to a cost function based on $L_{X_S}$, we obtain a desired dynamic, in which gates corresponding to features that contain high levels of noise gradually close, and gates corresponding to features that are consistent with the true structures in the data gradually become fully open. This is demonstrated in Fig. 3.

## 4.2 Stochastic Gates

Recently, several authors have incorporated continuous approximations of discrete random variables into neural network training [22, 16]. Such relaxations have been used for many applications, such as

---

**Algorithm 1** Differentiable Unsupervised Feature Selection (DUFS) Pseudo-code

---

**Input:** data, $\{\boldsymbol{x}_1, \ldots, \boldsymbol{x}_n\} \subset \mathbb{R}^d$, regularization parameter $\lambda$, required number of features $s$, number of epochs $T$.
Initialize the gate parameters: $\mu_i = 0.5$ for $i = 1, \ldots, d$.
**for** $t = 1$ **to** $T$ **do**
    Sample a stochastic gate (STG) vector $\boldsymbol{Z} = (Z_1, \ldots, Z_d)$ as described in (6).
    Apply the STG to the data $\boldsymbol{x}_i$ to obtain the gated data $\tilde{\boldsymbol{x}}_i$ ( $\tilde{\boldsymbol{x}}_i = \boldsymbol{x}_i \odot \boldsymbol{Z}$).
    Compute a kernel matrix $\boldsymbol{K} \in \mathbb{R}^{n \times n}$ (see (1)) using the gated data $\{\tilde{\boldsymbol{x}}_i\}$.
    Compute the graph Laplacian $\boldsymbol{L}_{\text{diff}} \in \mathbb{R}^{n \times n}$ as described in (2).
    Update $\mu_1, \ldots, \mu_d$ by applying GD to the loss function ((8) or (9)).
**end for**
Return $s$ features with largest $\mu_i$.

---

model compression [21], discrete softmax activations [15], and feature selection [36]. Here, we use a Gaussian-based relaxation of Bernoulli variables, termed Stochastic Gates (STG) [36], which relies on the repamaterization trick [23, 11], to reduce the variance of the gradient estimates.

We denote the STG random vector as $\boldsymbol{Z} \in [0,1]^d$, parametrized by $\mu \in \mathbb{R}^d$. Each vector entry is defined as

$$Z_i = \max(0, \min(1, \mu_i + \epsilon_i)), \tag{6}$$

where $\mu_i$ is a learnable parameter, $\epsilon_i$ is drawn from $\mathcal{N}(0, \sigma^2)$, and $\sigma$ is fixed throughout training. This approximation can be viewed as a clipped, mean-shifted, Gaussian random variable. In Fig. 4, we illustrate the gating mechanism and show examples of the densities of $Z_i$ for different values of $\mu_i$. Note that, even though $Z_i$ is not differentiable at 0 or 1, we can use the sub/sup-gradient, which is defined using the one-sided gradient of the function at 0 or 1.

Multiplication of each feature by its corresponding gate enables us to derive a fully differentiable feature selection method. At initialization $\mu_i = 0.5$ for $i = 1, ..., d$, so that all gates approximate a "fair" Bernoulli variable. The parameters $\mu_i$ can be learned via gradient decent by incorporating the gates into a diffrentiable loss term. To encourage feature selection in the supervised setting, [36] proposed the following differentiable regularization term:

$$r(\boldsymbol{Z}) = \sum_{i=1}^{d} \mathbb{P}(Z_i > 0) = \sum_{i=1}^{d} \left( \frac{1}{2} - \frac{1}{2} \operatorname{erf}\left(-\frac{\mu_i}{\sqrt{2}\sigma}\right) \right), \tag{7}$$

where $\operatorname{erf}()$ is the Gauss error function. The term (7) penalizes open gates, encouraging those corresponding to features that are not useful for prediction to transition into a closed state (which is the case for small $\mu_i$).

### 4.3 Differentiable Unsupervised Feature Selection (DUFS)

Let $\boldsymbol{X} \in \mathbb{R}^{m \times d}$ be a data mini-batch. Let $\boldsymbol{Z} \in [0,1]^d$ be a random variable representing the stochastic gates, parametrized by $\boldsymbol{\mu} \in \mathbb{R}^d$, as defined in Section 2. For each mini-batch, we draw a vector, $\mathbf{z}$,

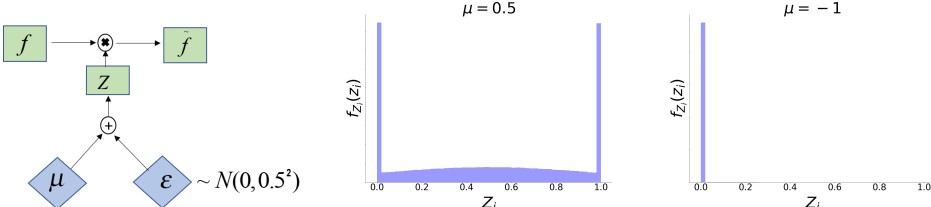

Figure 4: Left: The stochastic gate $Z$ is defined via the repamaterization trick [23, 11]. Standard Gaussian noise is injected and shifted by a trainable parameter $\mu_i$; the sum is thresholded to $[0,1]$ based on (6). Two examples of the density of the stochastic gate $Z_i$. Middle: at initialization, $\mu_i = 0.5$, and the gate approximates a 'fair' Bernoulli variable. Right: the distribution at $\mu_i = -1$ approximates a 'closed' gate.

of realizations from $\boldsymbol{Z}$ and define a matrix, $\tilde{\boldsymbol{Z}} \in [0, 1]^{m \times d}$, consisting of $m$ copies of $\mathbf{z}$. We denote $\tilde{\boldsymbol{X}} \triangleq \boldsymbol{X} \odot \tilde{\boldsymbol{Z}}$ as the gated input, where $\odot$ is an element-wise multiplication, also known as the Hadamard product. Let $\boldsymbol{L}_{\tilde{X}}$ be the random walk graph Laplacian computed on $\tilde{\boldsymbol{X}}$ .

We propose two loss function variants. Both variants contain a feature scoring term, $-\frac{1}{m}\mathrm{Tr}[\tilde{\boldsymbol{X}}^T \boldsymbol{L}_{\tilde{X}} \tilde{\boldsymbol{X}}]$, and a feature selection regularization term, $\sum_{i=1}^{d} \mathbb{P}(\boldsymbol{Z}_i \geq 0)$, as in (7). In the first variant (8), the two terms are balanced using a hyperparameter $\lambda \geq 0$.

$$L(\boldsymbol{\mu};\ \lambda) := -\frac{\mathrm{Tr}\big[\tilde{\boldsymbol{X}}^T \boldsymbol{L}_{\tilde{X}} \tilde{\boldsymbol{X}}\big]}{m} + \lambda \sum_{i=1}^{d} \mathbb{P}(\boldsymbol{Z}_i \geq 0). \tag{8}$$

Tuning $\lambda$ allows for flexibility in the number of selected features. In Section S2 of the Appendix, we present an unsupervised scheme for tuning $\lambda$. To obviate the need to tune $\lambda$, we introduce an alternative parameter-free loss function:

$$L_{\text{param-free}}(\boldsymbol{\mu}) := -\frac{\mathrm{Tr}\big[\tilde{\boldsymbol{X}}^T \boldsymbol{L}_{\tilde{X}} \tilde{\boldsymbol{X}}\big]}{m \sum_{i=1}^{d} \mathbb{P}(\boldsymbol{Z}_i \geq 0) + \delta} \ , \tag{9}$$

where $\delta$ is a small constant added to circumvent division by $0$. The parameter-free variant (9) seeks to minimize the average score per selected feature, where the average is calculated as the total score (in the numerator) divided by a proxy for the number of selected features (the denominator). Minimizing both objectives (8) and (9) will encourage the gates to remain open for features that yield high Laplacian scores and closed for the remaining features.

Our algorithm involves applying a standard optimization scheme (such as stochastic gradient decent) to objective (8) or (9). This optimization algorithm will use the following calculation.

$$\frac{d}{d\mu_i}Tr[\tilde{\boldsymbol{X}}^T \boldsymbol{L}_{\tilde{X}} \tilde{\boldsymbol{X}}] = \begin{cases} 2\boldsymbol{Z}_i(\tilde{\boldsymbol{X}}_{:,i})^T \boldsymbol{L}_{\tilde{X}} \tilde{\boldsymbol{X}}_{:,i} + \sum_{s,t}(\tilde{\boldsymbol{X}}\tilde{\boldsymbol{X}}^T)_{s,t}\frac{d}{d\mu_i}\left(\frac{K_{t,s}}{\sum_l K_{t,l}}\right) & \boldsymbol{Z}_i \in (0,1) \\ 0 & \boldsymbol{Z}_i \in \{0,1\} \end{cases}$$

$$\frac{d}{d\mu_i}\sum_{i=1}^{d} \mathbb{P}(\boldsymbol{Z}_i \geq 0) = \frac{d}{d\mu_i}\,\mathrm{erf}\left(\frac{\mu_i}{\sqrt{2}\sigma}\right),$$

where $\mathrm{erf}()$ is the Gauss error function and $\tilde{\boldsymbol{X}}_{:,i}$ denotes the $i-$th column of $\tilde{\boldsymbol{X}}$. After training, we remove the stochasticity ($\epsilon_i$ in (6)) from the gates and retain features satisfying $Z_i > 0$. At each step of the training procedure, we compute a kernel $\boldsymbol{K}$, (see (1)), based on a mini-batch of size $m$. The complexity of this calculation is $\mathcal{O}(mk^2)$ [7], using the top $k$ nearest neighbors for each point. A pseudo code describing the steps of our method is shown in Algorithm 1.

### 4.3.1 Raising $L$ to the $t$'th Power

Replacing the Laplacian $\boldsymbol{L}$ in equations (8) and (9) by its $t$-th power, $\boldsymbol{L}^t$, with $t > 1$ corresponds to taking $t$ random walk steps [24]. This suppresses the smallest eigenvalues of the Laplacian, while preserving its eigenvectors. We used $t = 2$, which was observed to improve the performance of our proposed approach (see Appendix for additional details).

## 5 Experiments

To demonstrate the capabilities of DUFS, we begin by presenting an artificial two-moons experiment. We then report results obtained on several standard datasets and compare them to existing unsupervised feature selection algorithms. When applying the method to real data we perform feature selection based on Eq. (8) using several values of $\lambda$. In the Appendix (Section S2), we describe a procedure for choosing the optimal value of $\lambda$. Next, following the analysis in [34], we perform $k$-means clustering using the leading $50, 100, 150, 200, 250$, or $300$ selected features and average the results over 20 runs. Leading features are identified by sorting the gates based on $\mathbb{P}(Z_i)$ (see (7)). The number clusters $k$ is set as the number of classes, and the sample labels are utilized to evaluate clustering accuracy. The best average clustering accuracy is recorded along with the number of selected features $|\mathcal{S}|$.

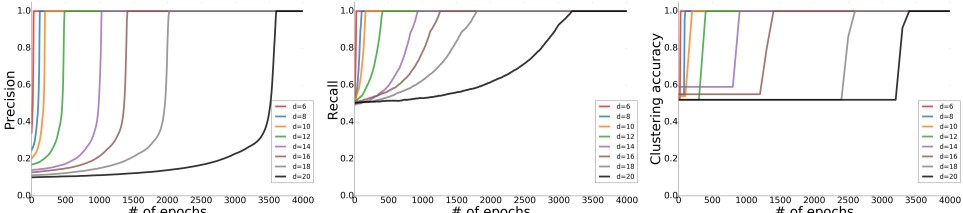

Figure 5: Evaluating the precision and recall of feature selection, in addition to clustering quality as a function of epoch number. We apply the parameter-free loss variant (see (9)) on the noisy two-moons data with a total of $d$ features, out of which only the leading 2 capture the structure of the two-moons. Left: feature selection precision. Here, precision is defined as the ratio between amount of retrieved informative features and all retrieved features, that is: $\frac{\sum_{i=1}^{2} P(Z_i > 0)}{\sum_{i=1}^{d} P(Z_i > 0)}$. Middle: feature selection recall. Here, recall is defined as the ratio between amount of retrieved informative features and all informative features, that is: $\frac{\sum_{i=1}^{2} P(Z_i > 0)}{2}$. Note that, in all of these examples, the gates converge to "deterministic" values. Namely, $P(Z_i > 0) \simeq 1$ for informative features $i = 1, 2$, and $P(Z_i > 0) \simeq 0$ for the nuisance features $i = 3, ..., d$. Right: clustering accuracy obtained with the retrieved features (recorded every 10 epochs). Here, clustering is performed using spectral clustering [26] with a Gaussian kernel.

## 5.1 Noisy Two-Moons

In the first experiment, we use a two-moons-shaped dataset (see Fig. 1) concatenated with nuisance features. The first two coordinates $\boldsymbol{f}_1, \boldsymbol{f}_2$ are generated by adding a Gaussian noise, with zero mean and variance of $\sigma_r^2 = 0.1$, to two nested half circles. Nuisance features $\boldsymbol{f}_i, i = 3, ..., d$, are drawn from a multivariate Gaussian distribution with zero mean and identity covariance. The total number of samples is $n = 100$. Note that the small sample size makes the task of identifying nuisance features more challenging.

We evaluate the convergence of the parameter-free loss (9) using gradient decent. We use a different number of features $d$ and plot the precision and recall of feature selection throughout training (see Fig. 5). In all of the presented examples, perfect precision and recall are achieved at convergence.

## 5.2 Noisy Image Data

In the following experiment, we evaluate our method on two noisy image datasets. The first is a noisy variant of MNIST [19], in which each background pixel is replaced by a random value drawn uniformly from $[0, 1]$ (see also [29]). Here, we focus only on the digits '3' and '8'. The second dataset is a noisy variant of PIXRAW10P (abbreviated PIX10), created by adding noise drawn uniformly from $[0, 0.3]$ to all pixels. In both datasets, the images were scaled into $[0, 1]$ prior to the addition of noise. We applied DUFS and LS to both datasets, identifying low frequency features. In the top panels of Fig. 6, we present the leading 50 features retained on the noisy MNIST along with the average clustering accuracy over 20 runs of $k$-means. In this case, DUFS' open gates concentrate at the left side of the handwriting area, which is the side that distinguishes '3' from '8'. This allows DUFS to achieve a higher clustering accuracy comparing to LS. The bottom panels of Fig. 6 show the leading 300 features retained on noisy PIX10 along with the average clustering accuracy. Here, DUFS selects features which are more informative for clustering the face images. We refer the reader to Appendix S4 for additional information on this experiment and for extended results on COIL20 and COIL100 [25].

## 5.3 Clustering of Real World Data

Here, we evaluate the capabilities of the proposed approach (DUFS) on real-world high dimensional datasets whose properties are summarized in Table 1 [3]. We compare DUFS to Laplacian Score (LS) [4], Multi-Cluster Feature Selection (MCFS) [6], Local Learning based Clustering (LLCFS) [37], Nonnegative Discriminative Feature Selection (NDFS) [20], Multi-Subspace Randomization and

---

[3]All datasets are publicly available, see description in Appendix section S7

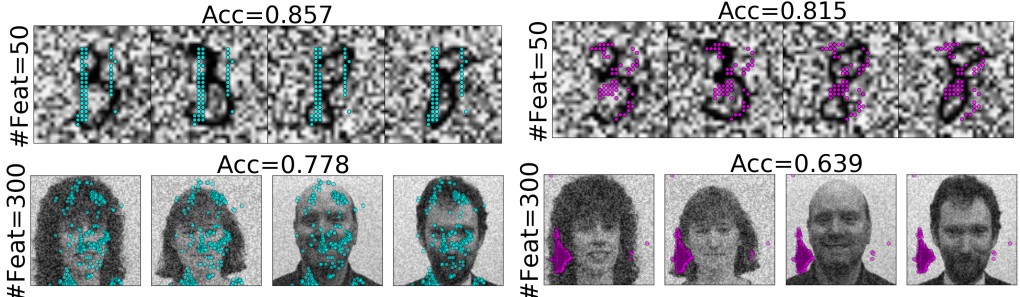

Figure 6: Noisy image experiments. Top: examples of noisy MNIST digits highlighted with the leading 50 features selected by DUFS (left) and LS (right). Bottom: examples from the noisy PIX10 datasets overlaid with the leading 300 features selected by DUFS (left) and LS (right). This figure is best viewed in color. The grayscale of MNIST images is inverted to improve visibility.

Collaboration (SRCFS) [14], and Concrete Auto-encoders (CAE) [1]. We compare the accuracy of clustering based on the feature selected by DUFS to those selected by the 6 baselines, and all features (All). As can be seen in table 1, DUFS outperforms all baselines on 9 datasets and ranks second on the remaining 3. Overall, the median and mean rankings of DUFS are 1 and 1.31, respectively. In the Appendix, we present a complementary table with the standard deviations of the clustering accuracies obtained in this example. Our results demonstrate that our method is extremely useful in bioinformatics; for example, when scientists study single-cell RNA sequencing (scRNA-seq), accurate cluster assignments are vital for downstream analysis. In table 1, the bottom 4 datasets were sequenced using scRNA-seq technology; on these datasets, DUFS improves the cluster assignments by 23.0% (on average) compared with assignments obtained based on all features.

| Datasets | LS [4] | MCFS [6] | NDFS [20] | LLCFS [37] | SRCFS [14] | CAE [1] | DUFS | All | Dim/Samples/Classes/Type |
|----------|--------|----------|-----------|------------|------------|---------|------|-----|--------------------------|
| GISETTE | 75.8 (50) | 56.5 (50) | 69.3 (250) | 72.5 (50) | 68.5 (50) | 77.3 (250) | **99.5 (50)** | 74.4 | 4955 / 6000 / 2 / Image |
| PIX10 | 76.6 (150) | 75.9 (200) | 76.7 (200) | 69.1 (300) | 75.9 (100) | **94.1 (250)** | 88.4 (50) | 74.3 | 10000 / 100 / 10 / Image |
| COIL20 | 60.0 (300) | 59.7 (250) | 60.1 (300) | 48.1 (300) | 59.9 (300) | 65.6 (200) | **65.8 (250)** | 53.6 | 1024 / 1444 / 20 /Image |
| Yale | 42.7 (300) | 41.7 (300) | 42.5 (300) | 42.6 (300) | 46.3 (250) | 45.4 (250) | **47.9 (200)** | 38.3 | 1024 / 165 / 15 /Image |
| RCV1 | 54.9 (300) | 50.1 (150) | 55.1 (150) | 55.0 (300) | 53.7 (300) | 54.9 (300) | **62.2 (300)** | 50.0 | 24408 / 21232 / 2 / Text |
| TOX-171 | 47.5 (200) | 42.5 (100) | 46.1 (100) | 46.7 (250) | 45.8 (150) | 47.7 (100) | **49.1 (50)** | 41.5 | 5748 / 171 / 4 / Bio |
| ALLAML | 73.2 (150) | 72.9 (250) | 72.2 (100) | **77.8 (50)** | 67.7 (250) | 73.5 (250) | 74.5 (100) | 67.3 | 7192 / 72 / 2 / Bio |
| PROSTATE | 56.8 (300) | 57.3 (300) | 58.3 (100) | 57.8 (50) | 60.6 (50) | 56.9 (250) | **64.7 (150)** | 58.1 | 5966 / 102 / 2 / Bio |
| SRBCT | 41.1(300) | 43.7(250) | 41.0(50) | 34.6(150) | 33.49(50) | **62.6 (200)** | 51.7 (50) | 39.6 | 2308 / 83 / 4 / Bio |
| BIASE | 83.8 (200) | 95.5 (300) | 100 (100) | 52.2 (300) | 50.8 (50) | 85.1 (250) | **100 (50)** | 41.8 | 25683 / 56 / 4 / Bio |
| INTESTINE | 43.2 (300) | 48.2 (300) | 42.3 (100) | 63.3 (200) | 58.1 (300) | 51.9 (50) | **71.9 (250)** | 54.8 | 3775 / 238 / 13 / Bio |
| FAN | 42.9 (150) | 45.5 (150) | 48.8 (100) | 29.0 (50) | 29.0 (100) | 35.2 (300) | **49.0 (50)** | 37.5 | 25683 / 56 / 8 / Bio |
| POLLEN | 46.9 (150) | **66.5 (300)** | 48.9 (50) | 35.0 (100) | 34.9 (300) | 58.0 (250) | 60.2 (50) | 54.9 | 21810 / 301 / 4 / Bio |
| Mean rank | 4.0 | 5.0 | 4.08 | 4.77 | 5.31 | 3.23 | **1.31** | | |
| Median rank | 4 | 5 | 4 | 5 | 6 | 3 | **1** | | |

Table 1: Left sub-table: Average clustering accuracy on several benchmark datasets. Clustering is performed by applying $k$-means to the features selected by the different methods. The number of selected features is shown in parenthesis. Right sub-table: Properties of the real world data used for empirical evaluation.

# 6 Conclusions

In this paper, we propose DUFS, a novel unsupervised feature selection method that introduces learnable Bernoulli gates into a Laplacian score. DUFS has an advantage over the standard Laplacian score, as it re-evaluates the graph Laplacian based on the subset of selected features. We demonstrate that our proposed approach captures structures in the data that are not detected by the standard Laplacian score. Finally, we experimentally demonstrate that our method outperforms current unsupervised feature selection baselines on several real-world datasets.

# Acknowledgements

The authors thank Stefan Steinerberger, Boaz Nadler and Ronen Basri for helpful discussions. The work of OL and YK was supported by the National Institutes of Health R01GM131642, UM1PA05141, P50CA121974, and U01DA053628.

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
