# Differentiable Unsupervised Feature Selection based on a Gated Laplacian - Supplementary Materials

October 25, 2021

## Contents

# S1 Tuning the Kernel's Bandwidth

It is important to properly tune the kernel scale/bandwidth $\sigma_b$, which determines its scale of connectivity. Several studies have proposed schemes for tuning $\sigma_b$, see for example [10, 3, 12, 5]. Here, we focus on two schemes, a global bandwidth and a local bandwidth. The local bandwidth proposed in [12], involves setting a local-scale $\sigma_i$ for each data point $\boldsymbol{x}_i, i = 1, ..., n$. The scale is chosen using the $L_1$ distance from the $k$-th nearest neighbor of the point $\boldsymbol{x}_i$. Explicitly, the calculation for each point is

$$\sigma_i = C \cdot ||\boldsymbol{x}_i - \boldsymbol{x}_k||^2, i = 1, ..., N, \tag{1}$$

where $\boldsymbol{x}_k$ is the $k$-th nearest (Euclidean) neighbor of the point $\boldsymbol{x}_i$, and $C$ is a predefined constant in the range $[1, 5]$. A global scale $\hat{\sigma}_b$, is defined as the max over $\sigma_i$. Using this value the kernel values are defined by

$$K_{i,j} = \exp\left(-\frac{||\boldsymbol{x}_i - \boldsymbol{x}_j||^2}{\hat{\sigma}_b}\right), i, j \in \{1 \dots n\}. \tag{2}$$

This scale guarantees that all of the points are connected to at least $k$ neighbors.

# S2 Tuning the Regularization Parameter

The regularization parameter $\lambda$ controls the amount of sparsity obtained by DUFS. A larger $\lambda$ would lead to a sparser solution in an earlier stage of training. If the number of desired features is prescribed (say $s$), $\lambda$ can be tuned to a value resulting in a solution sparsity close to $s$. However, this procedure might often be time consuming. Alternatively, we propose here a "warm-up" procedure (see Algorithm 1) for evaluating the optimal choice of $\lambda$ in DUFS.

In Algorithm 1, we propose to first set a grid of hypothesis values, for instance $\lambda \in \{0.01, 0.1, 1, 10, 100\}$, then run DUFS with $n_{\text{epochs}} = 1,000$ for each value of $\lambda$. For each value we sort the gates and select the leading $s$ features with the largest gates coefficients. For each $\lambda$ we denote the dataset restricted to the leading $s$ features as $\boldsymbol{X}_\lambda$. We then evaluate each selected set by the feature scoring term $S(\lambda) = \frac{1}{s}\text{Tr}[\boldsymbol{X}_\lambda^T \boldsymbol{L}_\lambda^2 \boldsymbol{X}_\lambda]$. We return the $\lambda$ values which maximizes this score. This $\lambda$ could be used to continue the training procedure for additional iterations.

---

**Algorithm 1** Warm Start Pseudo-Code

---

**Input:** data $\{\boldsymbol{x}_1, \dots, \boldsymbol{x}_n\} \subset \mathbb{R}^d$, required number of features $s$, (the set of $\lambda$ values can be input as well).

**for** $\lambda \in \{0.01, 0.1, 1, 10\}$ **do**

Run DUFS (Algorithm 1) on the data with $\lambda$, for $T = 1,000$ epochs, and get the leading $s$ features.

Define $\boldsymbol{X}_\lambda$ to be the dataset restricted to the $s$ leading features.

Compute the graph Laplacian $L_\lambda \in \mathbb{R}^{n \times n}$ as described in (2).

Evaluate the Laplacian Score for the leading $s$ features

$$S(\lambda) = \frac{1}{s}Tr[\boldsymbol{X}_\lambda^T \boldsymbol{L}_\lambda^2 \boldsymbol{X}_\lambda]$$

**end for**

Return $\lambda$ with the largest $S(\lambda)$

---

## S3  Additional Experimental Details

In the following sections we provide additional experimental details required for reproduction of the experiments provided in the main text. All the experiments are conducted using Intel(R) Xeon(R) CPU E5-2620 v3 @2.4Ghz x2 (12 cores total).

## S4  Feature selection on Image Datasets

Here, we provide a deeper look into the features identified by the proposed method when applied to image data. We start with COIL20[7] which is a data that contains 20 objects captured at different viewing angles. In Fig. S1 we present the leading $\{50, 100, ..., 300\}$ features selected by DUFS and LS along with the average clustering accuaracies based on the selected features. In this example DUFS selects features which lie on the symmetry axis of COIL20, these features are more informative for clustering COIL20 since the values of rotated objects vary slowly on this axis. Next, we present a similar comparison on COIL100[7]. COIL100 contains 7200 samples of 100 objects captured at different angles. Each image is of dimension $[128, 128, 3]$. In Fig. S2 we present the leading $\{50, 100, ..., 300\}$ features selected by DUFS and LS along with the average clustering accuracies based on the selected features. Here, feature selection is performed based on a black and white version of the RGB image and clustering is performed based on the corresponding subset of pixels from the RGB tensor.

Finally, in Fig. S3 we present the results of application of DUFS to the noisy MNIST dataset. This is an extension of the results presented in the paper. Specifically, we demonstrate the clustering accuracies based on the leading $\{50, 100, ..., 300\}$ features selected by DUFS and LS. In this experiment, we focused on a random subset of 1000 samples of the digits 3 and 8.

## S5  Raising $L$ to the $t$'th Power

To suppress the smallest eigenvalues of the Laplacian, we have suggested to replace the Laplacian $L$ in equations (8) and (9) by its $t$-th power $L^t$ with $t > 1$. As shown in [6] this corresponds for taking $t$ random walk steps on the graph of the data. In this subsection we empirically demonstrate the effect of $t$ using the two-moons dataset (described in the Experimental section of the main text). We construct the two-moons dataset with different number of nuisance variables ($d$) and apply DUFS (with the parameter free loss) computed based on $L$ raised to the power of $t = 1, 2$ and 3. In Fig. S4 we present the clustering accuracy (averaged over 100 runs) based on $k$-means, which is applied to the selected features. As evident in this plot the Laplacian based on $t = 2$ yields better performance for a wider range of nuisance variables in this experiment. Following this result, we keep $t = 2$ in all of our examples.

## S6  Extended Clustering Results

In the next experiment, we evaluate the effectiveness of the proposed method for different numbers of selected features on 3 datasets. We compare DUFS versus LS by performing $k$-means clustering using the features selected by each method. In Fig. S5, we present the clustering accuracies (averaged over 20 runs) based on the leading $\{50, 100, ..., 300\}$ features. We see that DUFS consistently selects features which provide higher clustering capabilities compared to LS.

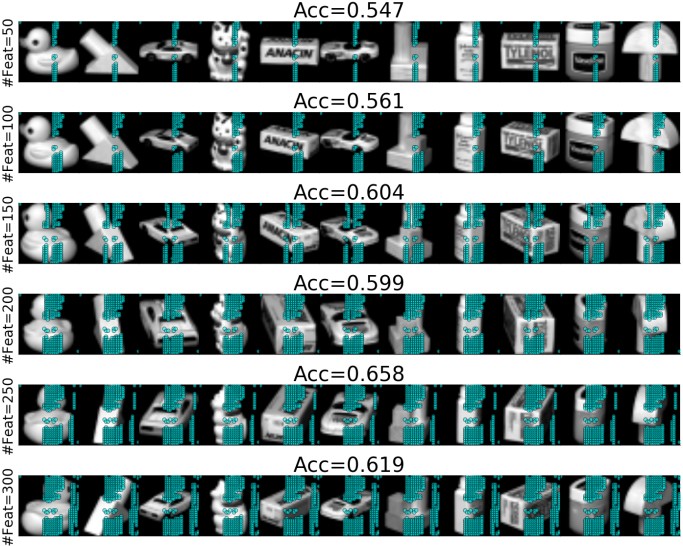

(a) DUFS

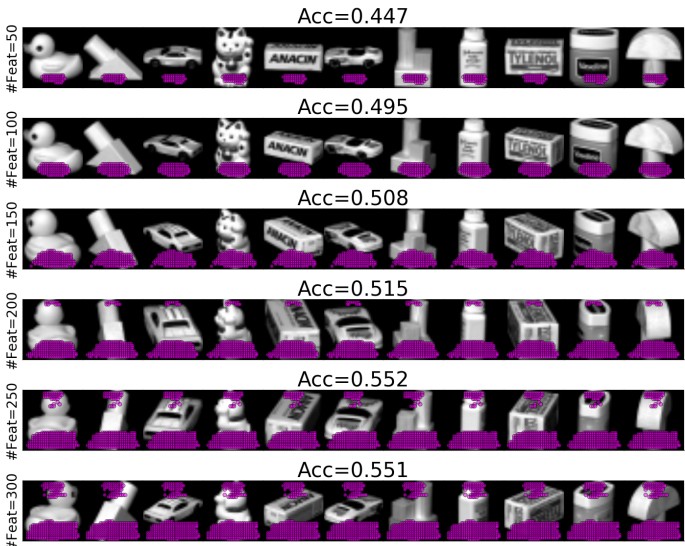

(b) LS

Figure S1: Features selected by DUFS and LS in the COIL20 dataset. Top: selected features (cyan dots) and clustering accuracy based on DUFS. Note that as COIL20 contains different angles of each object, the selected feature lie approximately on the symmetry axis. Bottom: selected features (magenta dots) and clustering accuracy based on LS.

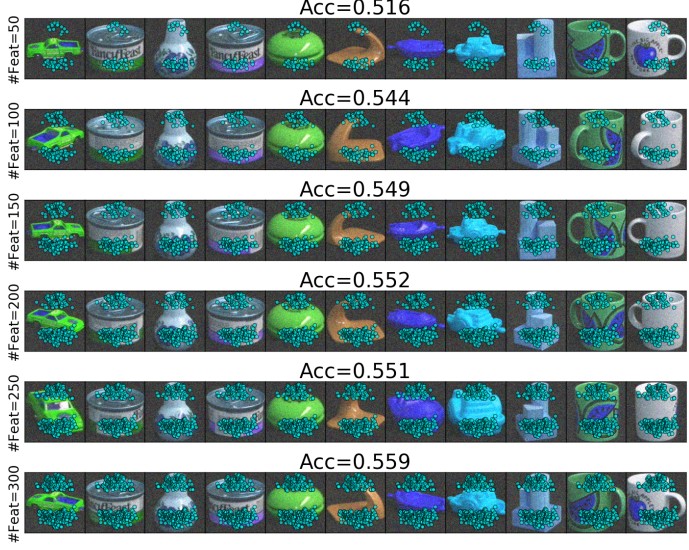

(a) DUFS

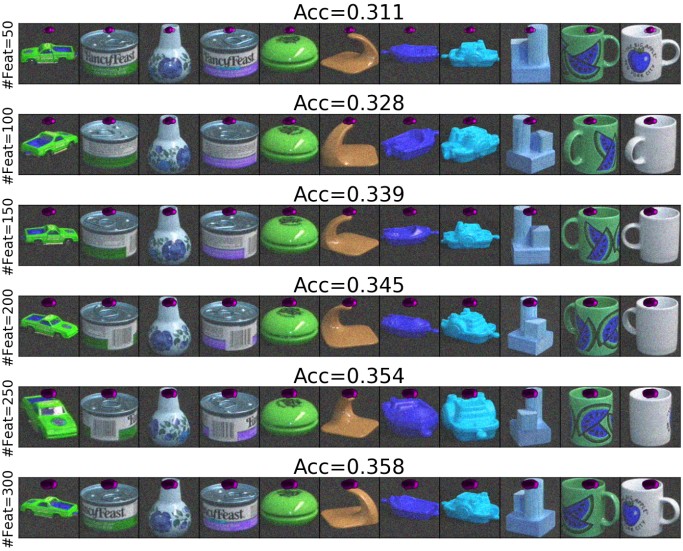

(b) LS

Figure S2: Same as for S1 but for the COIL100 dataset. Note that as COIL100 contains different angles of each object, the selected feature lie approximately on the symmetry axis. In this example, the LS also selects features on the symmetry axis, however the LS based selected features are condensed at a small region near the top part of the image. These features are informative for clustering wide vs. long objects but less informative for clustering all 100 objects.

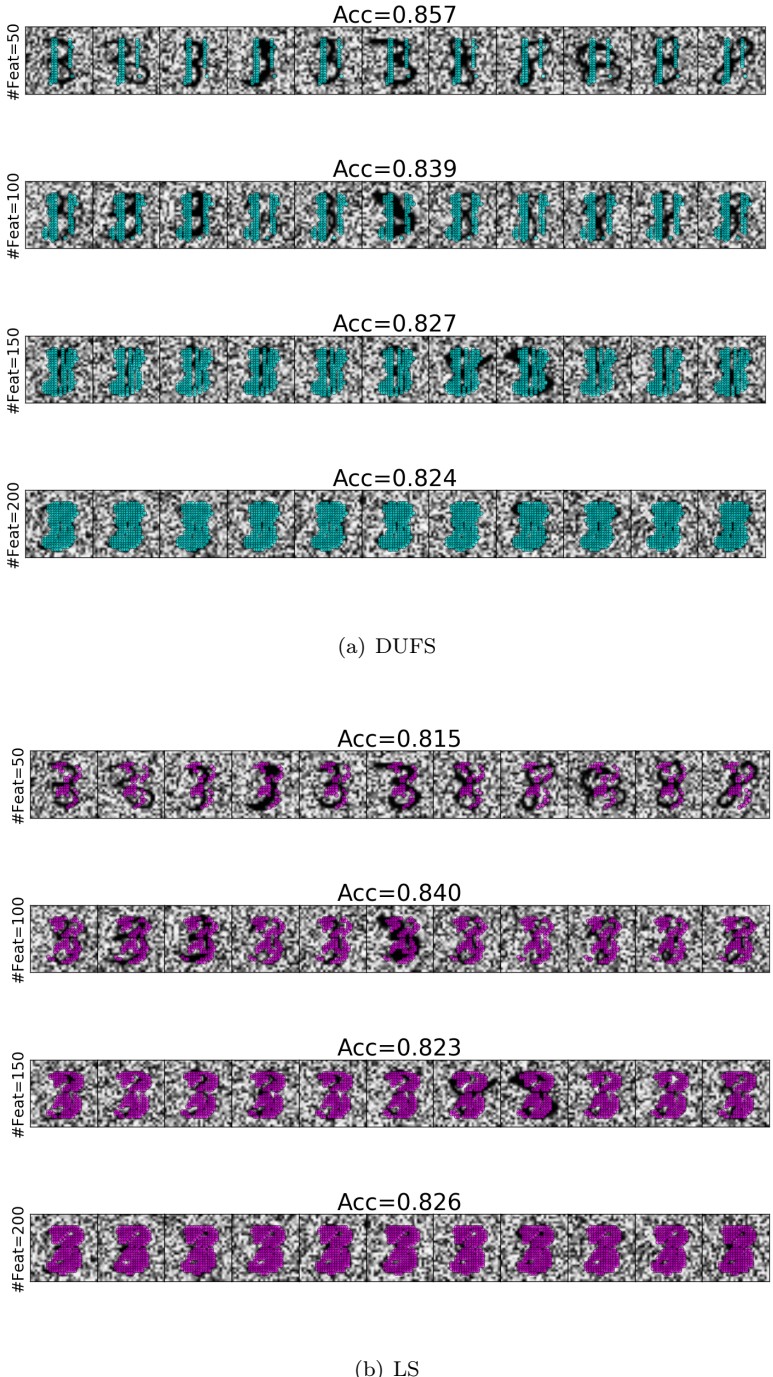

(a) DUFS

(b) LS

Figure S3: Selected features on MNIST dataset. Top: selected features and clustering accuracy based on DUFS. Bottom: selected features and clustering accuracy based on LS. In this example, DUFS outperforms the LS when it is regularized to select a small number of features. However, when the regularization is set to select > 150 features in DUFS, the features with top scores in both methods are similar.

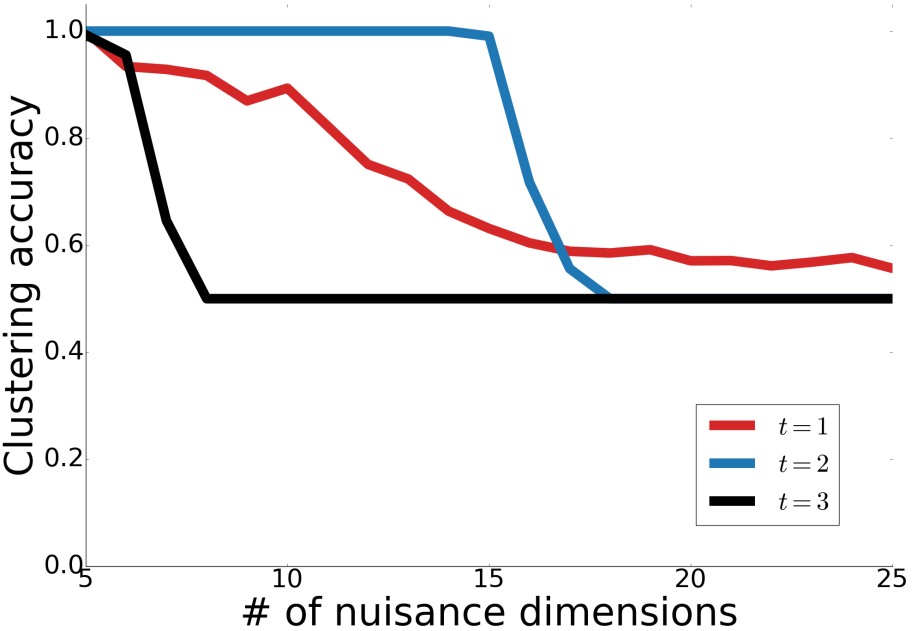

Figure S4: Taking powers of the graph Laplacian. Clustering accuracy vs. number of nuisance dimensions in the two-moons datasts. We apply the parameter free variant of DUFS using a Laplacian $\boldsymbol{L}$ raised to the power of $t$. Clustering is performed using $k$-means applied to the selected features and averaged over 100 runs of DUFS.

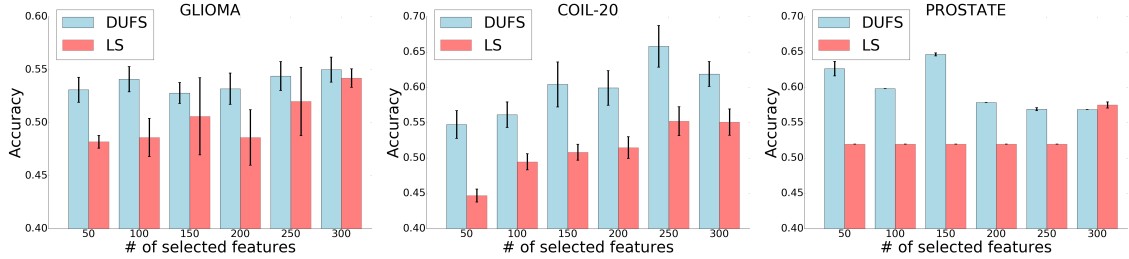

Figure S5: Clustering accuracy on three real world datasets. Clustering was performed by applying $k$-means to features selected by DUFS and LS. The averages and standard deviations based on 20 runs are shown.

## S7    Experimental Details

In this subsection we describe all required details for conducting the experiments provided in the main text. The datasets are all publicly available and can be downloaded from [1],[2],[3]. The scRNA-seq datasets were collected by the following authors [1, 9, 2, 8].

We use SGD for all the experiments which are conducted using Intel(R) Xeon(R) CPU E5-2620 v3 @2.4Ghz x2 (12 cores total). For LS, MCFS, and NDFS we use a python implementation from [4]. For LLCFS and SRCFS we use a Matlab implementation from [5] and [6]. For CAE we use a python implementation available at [7]. For DUFS and LS, we use $k = 2$ (number of nearest neighbors) which worked well on all of the datastes, except for the GISSETE dataset in which we used $k = 5$. The factor $C$ (see Eq. 1) in all experiments is 5 except for SRBCT, COIL20 and PIX10 in which $C = 2$. All datasets are publicly available at [8], except RCV1 which is available at [9]. RCV1 is a multi-class multi-label datasets, in our analysis we use a binary subset of RCV1. To create this subset, we focus on the first two classes and remove all samples that have multiple labels, then we balance the classes by down sampling the larger class. For NDFS, MCFS and SRCFS we use $k = 5$ for the affinity matrix $\boldsymbol{W}$, note that NDFS and LLCFS use the number of clusters for selecting features. The tuning process for hyper-parameters of all method follows the grid search described described in [11].

In all examples except RCV1, COIL100 and COIL20 we use a full batch size for computing the kernel, for COIL100 and COIL20 the batch size is 1000. For all two-moons examples presented in Fig. 1 we use the parameter free loss term with a learning rate (LR) of 1 and 5000 epochs. For PROSTATE data, we use a learning rate of 1, 12000 epochs and $\lambda$ is evaluated in the range $[0.01, 1]$. For GLIOMA data we use a learning rate of 0.3, 12000 epochs and $\lambda$ is evaluated in the range $[3, 30]$. For ALLAML data we use a learning rate of 0.3, 20000 epochs and $\lambda$ is evaluated in the range $[1, 5]$. For COIL20 data we use a learning rate of 0.3, 26000 epochs and $\lambda$ is evaluated in the range $[0.01, 2]$. For COIL100 data we use a learning rate of 1, 6000 epochs and $\lambda$ is evaluated in the range $[0.01, 2]$. For PIX10 data we use a learning rate of 0.3, 20000 epochs and $\lambda$ is evaluated in the range $[0.05, 1]$. For SRBCT, BIASE, INTESTINE, FAN, and Pollen datasets we use a learning rate of 0.3, 20000 epochs and $\lambda$ is evaluated $[0.001, 0.1]$.

## S8    Strengths and Limitations

The proposed method provides several advantages compared to the classic Laplacian Score: (1) it sparsifies the Laplacian and therefore, can identify subsets of low-frequency feature even in the presence of a large number of nuisance variables, (2) it can be computed in small batches, therefore is computationally lighter than the Laplacian Score. The success of our method relies on the assumption that the data contains nuisance variables that are of high-frequency nature. However, often real datasets contain variables that are weakly correlated with the structure of the data. In such cases, DUFS may capture these weakly correlated variables even though removing them might be beneficial for downstream analysis tasks. In the future, we plan to extend DUFS to handle correlated variables.

---

[1]http://featureselection.asu.edu/datasets.php
[2]We use a binary subset of RCV1
[3]For SRBCT we removed 4 samples that had 3 labels. The data can be found on [4]
[4]https://github.com/jundongl/scikit-feature
[5]https://github.com/huangdonghere/SRCFS
[6]https://www.mathworks.com/matlabcentral/fileexchange/56937-feature-selection-library
[7]https://github.com/mfbalin/Concrete-Autoencoders
[8]http://featureselection.asu.edu/datasets.php
[9]https://scikit-learn.org/0.18/datasets/rcv1.html

| Datasets | LS | MCFS | NDFS | LLCFS | SRCFS | CAE | DUFS | All | Dim/Samples/Classes/Type |
|---|---|---|---|---|---|---|---|---|---|
| GISETTE | 75.8 (1.9E-2) | 56.5 (2.2E-14) | 69.3 (7.5E-1) | 72.5 (2.2E-2) | 68.5 (2.7E-1) | 77.3 (2E-2) | **99.5 (1.1E-14)** | 74.4 (3.3E-1) | 4955 / 6000 / 2 / Image |
| PIX10 | 76.6 (8.1) | 75.9 (8.59) | 76.7 (8.52) | 69.1 (4.5E-2) | 75.9 (7.0) | **94.1 (5.6E-1)** | 88.4 (3.7) | 74.3 (12.1) | 10000 / 100 / 10 / Image |
| COIL20 | 60.0 (3.3) | 59.7 (2.3) | 60.1 (1.6) | 48.1 (1.9E-2) | 59.9 (2.3E-2) | 65.6 (2.1) | **65.8 (2.6)** | 53.6 (1.9) | 1024 / 1444 / 20 /Image |
| Yale | 42.7 (2.6) | 41.7 (2.2) | 42.5 (1.6) | 42.6 (2E-2) | 46.3 (2.5) | 45.4 (3.4) | **47.9 (2.5)** | 38.3 (2.2) | 1024 / 165 / 15 /Image |
| RCV1 | 54.9 (2.1) | 50.1 (7.5E-2) | 55.1 (1.4E-2) | 55.0 (1.1E-14) | 53.7 (1.6E-2) | 54.9 (4.2) | **62.2 (11.1)** | 50.0 | 24408 / 21232 / 2 / Text |
| TOX-171 | 47.5 (7.6E-1) | 42.5 (2.6) | 46.1 (5.4E-1) | 46.7 (1.5) | 45.8 (5.7) | 47.7 (7.5E-1) | **49.1 (2.7)** | 41.5 (2.1) | 5748 / 171 / 4 / Bio |
| ALLAML | 73.2 (1.1E-14) | 72.9 (1.7) | 72.2 (1E-14) | **77.8 (3.3E-14)** | 67.7 (6.1) | 73.5 (3E-1) | 74.5 (6E-1) | 67.3 (3.1) | 7192 / 72 / 2 /Bio |
| PROSTATE | 58.6 (1.1E-14) | 57.3 (1.1E-14) | 58.3 (1.1E-14) | 57.8 (1.1E-14) | 60.6 (1.8) | 56.9 (4.6E-1) | **64.7 (2.14E-1)** | 58.1 (1.1E-14) | 5966 / 102 / 2 /Bio |
| SRBCT | 41.1(2.8) | 43.7(2.6) | 41(2.2) | 34.58(5.2) | 33.49(5.2) | **62.6 (7.3)** | 51.7 (0.5) | 39.6(2.8) | 2308 / 83 / 4 / Bio |
| BIASE | 83.8 (0.39) | 95.5 (3.3) | 100 (0) | 52.2 (5.1) | 50.8 (4.9) | 85.1 (1.6) | **100 (0)** | 41.8 (8.5) | 25683 / 56 / 4 / Bio |
| INTESTINE | 43.2 (3.5) | 48.2 (4.0) | 42.3 (2.0) | 63.3 (10.3) | 58.1 (6.9) | 51.9 (3.8) | **71.9 (6.9)** | 54.8 (2.6) | 3775 / 238 / 13 / Bio |
| FAN | 42.9 (0.86) | 45.5 (2.6) | 48.8 (1.0) | 29.0 (3.4) | 29.0 (3.1) | 35.2 (3.4) | **49.0 (1E-14)** | 37.5 (0.72) | 25683 / 56 / 8 / Bio |
| POLLEN | 46.9 (9.9E-2) | **66.5 (3.0)** | 48.9 (3.6) | 35.0 (5.3) | 34.9 (2.9) | 58.0 (4.3) | 60.2 (0.24) | 54.9 (5.6) | 21810 / 301 / 4 / Bio |

Table S1: Left sub-table- Average clustering accuracy on several benchmark datasets. Clustering is performed by applying $k$-means 20 times using the features selected by the different methods. The standard deviation is shown in parenthesis. Last column represents the clustering accuracies when using all features. Right sub-table- Properties of the real world data used for empirical evaluation.