# OpenReview forum: "Differentiable Unsupervised Feature Selection based on a Gated Laplacian"
_NeurIPS.cc/2021/Conference — NeurIPS 2021 Poster_

### Official Review · Reviewer_2vfK · 2021-07-16

**Rating:** 7
**Confidence:** 3

**Summary:**

This paper studies the problem of unsupervised feature selection for clustering. It proposes adding stochastic gates to the features such that those that minimize the Laplacian score are learned, and the algorithm iteratively updates the stochastic gate parameters and the random walk graph Laplacian matrix (that is computed only using the features selected by the gates). Experimental results on both simulated and real-world data (e.g. image data, single cell RNA-sequence data) suggest improvement over the naive Laplacian score and other unsupervised feature selection methods.

**Limitations And Societal Impact:**

Yes.

**Main Review:**

This paper demonstrates very strong experimental results, especially on clustering real-world datasets (Table 1). This seems different enough from previous works, and has promise to be used by practictioners who work with clustering high-dimensional data (e.g. computational biologists). Also, the code and experimental details are provided in the supplementary material.

However, the paper could be improved by addressing the following questions:
- The existing methods that are compared in Section 5.3 are not the same as the prior work discussed in Section 1. How are these methods different from the proposed work/prior work and why were they chosen for comparison? How does the proposed work compare to other extensions of LS (mentioned in line 39 page 2)?
- What is the run time/actual computation time of the proposed algorithm? How does it compare to the other methods?
- In Section 5.3, is clustering done identically across the different feature selection methods, i.e. is the difference in performance solely due to the difference in feature selection? Is k-means clustering applied after DUFS?
- It's not clear to me how much Section 3, especially 3.2, adds to the paper. Why does the result from this exercise suggest that "Laplacian is sensitive to the number of nuisance dimensions in large datasets"?

The paper is easy to follow most of the time, but a few comments on clarity:
- In Algorithm 1, should $\tilde{x}_i = x_i \odot Z_i$? And is $L_{diff} = L_{\tilde{X}}$, given that $L_{diff}$ does not appear in (8) or (9)?
- There are some awkward margins, especially before and after figures. For example, Figure 3 cuts off the paragraph above.

**Time Spent Reviewing:**

3

---

> ### Author Response · Authors · 2021-08-10
> **Reply to Reviewer #4 - 2vfK**
>
> We would like to thank the Reviewer for the feedback, suggestions, and constructive comments. We provide our response below:
>
> 1. **Significance to practitioners** - Thanks, we believe that the method is beneficial for practitioners (e.g. computational biologists). In fact, we recently demonstrated that the proposed scheme can identify a subset of features related to the cell cycle in single-cell RNA-Seq data consisting of thousands of cells. Embedding the data using this subset of selected features (genes) reveals a circular manifold structure of the data points (cells included in the experiment). We further verify that the identified genes coincide with known markers for the cell cycle. Importantly, our method is able to discover these genes in a completely unsupervised fashion. We will elaborate on this result in the revised version.
>
> 2. **Selected baselines**- The baseline methods selected in Section 5.3 are relatively advanced (both in terms of performance and publication year) for unsupervised feature selection and are used as baselines in several recent publications such as [19,37]*. Other methods cited in the introduction (for example [27,38,39]*) were published more than a decade ago and do not have publically available code. Recent approaches that we use outperform these methods.
> *Citation numbers are for the references in the manuscript.
>
> 3. **Run time**- We describe the computational complexity involved in each iteration of the method in line 233, page 7. Specifically, each iteration requires O(mk^2) calculations, where m is the minibatch size, and k is the number of nearest neighbors used to construct the kernel. In terms of actual runtime, the method requires between minutes to hours on the evaluated real datasets. We will add a runtime benchmark experiment to the supplemental material.
>
> 4. **Clustering protocol** - Yes, the clustering performance difference is solely due to the difference in feature selection.  The clustering (using k-means) is performed after features are identified by DUFS or by any of the baseline methods. This means that improved clustering capabilities imply that the selected features better preserves the cluster structure.
>
> 5. **Clarification on the theoretical result Section 3**- One of the main contributions of our work is that we identify that the Laplacian Score (LS) breaks in the presence of many nuisance features. This limitation is proved theoretically in section 3.2, in which we demonstrate using an elementary setting (that enables theoretical analysis) that the  LS can not capture the smooth data structure in the presence of many nuisance features. This analysis motivates our work and encourages our designed solution to attenuate nuisance features before the computation of the LS. This leads to a clean, more robust LS that improves the performance on several real datasets.
>
> 6. **Typos**- Thanks for pointing out these typos and mistakes, corrected.
>
> **Summary** - We hope the response above clarifies several concerns and helps emphasize our novelty & merits to the reviewer. We would like to thank the reviewer again for acknowledging the significance of our work and for helping us improve the paper.

---

> > ### Comment · Reviewer_2vfK · 2021-09-10
> > **Acknowledgement**
> >
> > Thank you for your response. I have updated my review score.

---

### Official Review · Reviewer_X1KW · 2021-07-16

**Rating:** 5
**Confidence:** 4

**Summary:**

This paper proposes an unsupervised feature selection method that introduces learnable Bernoulli gates into a Laplacian score. Specifically, The proposed method utilizes stochastic input gates, trained to select features with high correlation with the leading eigenvectors of a graph Laplacian computed based on these features. Finally, compared with baselines, the proposed method can improve cluster assignments using some real datasets.

**Limitations And Societal Impact:**

The authors addressed the limitations and potential negative societal impact of their work. However, there are some concerns as follows:
1.The main concern is the innovation of this paper. Firstly,  laplacian score is proposed by Ref.13 for feature selection as an unsupervised measure. Secondly, i think that the main contribution of this paper is stochastic gates, but in Ref.36, the technology of stochastic gates is already used in supervised feature selection. Finally, authors focus on the traditional unsupervised feature selection problem. Thus i think that the core contribution of this paper is that authors extend the supervised problem in Ref.36 to the unsupervised problem without theoretical guarantees. Even authors introduce the importance of unsupervised feature selection from a diffusion perspective, but i don't think this is the core contribution of this article. For this question, if the authors can persuade me , I will change my score.
2.Authors introduce the importance of unsupervised feature selection from a diffusion perspective and i think this is a very novel thing for feature selection, but i can't understand what is the difference between similarity and exit times in nature. I hope the author can give me a more detailed explanation to understand the difference.
3.Authors sample a stochastic gate (STG) vector in algorithm 1 and thus i think that the proposed method should have randomness. But in the main experiment of this paper, i don't see this randomness analyzed by authors.
4.It would be better if the authors add some future work.

**Main Review:**

This paper is well organized and clearly written.
This paper provides enough information to reproduce its results.
The submission addresses a difficult task in a better way than previous work, but i think it is hard  to use the ideas or build on them.
The work is a combination of well-known techniques and related work is adequately cited, but i think it is not very clear how this work differs from previous contributions.
The proposed method is supported by sufficient experimental results without theoretical analysis.

**Time Spent Reviewing:**

8 hours

---

> ### Author Response · Authors · 2021-08-10
> **Reply to Reviewer #3 - X1KW**
>
> We would like to thank the Reviewer for the feedback, suggestions, and constructive comments. We provide our response below:
> 1. **Innovation** - Thanks, we believe that addressing this challenging important task is significant for practitioners.
> We agree with the reviewer that the method involves a combination of two existing methodologies. However, we argue that one of the main contributions of our work is the fundamental observation that the Laplacian Score (LS) breaks in the presence of many nuisance features. The performance break of the LS was demonstrated empirically in Figure 1 and then theoretically analyzed in a specific setting in Section 3.2. This limitation can explain the poor performance of the Laplacian score method (Ref. 13) on many real high-dimensional datasets (see Table 1). Then, to address the LS's current limitation, we propose adding a regularization term to enable re-evaluating the LS on a subset of features in a differentiable fashion. To the best of our knowledge, we are the first to propose regularizing the LS to enable sparsification of the features before the calculation of the LS.  Indeed our regularization term relies on a recently proposed non-convex regularization proposed for supervised learning. We argue that this combination is non-trivial, requires adaptation, and that combining existing techniques is part of the academic paradigm for advancing science. Our results demonstrate improvement over state-of-the-art unsupervised feature selection methods; this relies on extensive simulations valuable for the community. Another contribution is our new parameter-free loss which doesn't require tuning the number of selected features. In summary, the synergy between the LS and the stochastic gates techniques allows unraveling underlying clusters or manifolds that could not be detected otherwise. There are multiple examples in the field of DNN in which combinations of procedures serve as novel contributions for improved predictions.
> 2. **Cluster exist times** - Cluster exit time is long when the similarity within the cluster is high, and the similarity between clusters is low. Nuisance coordinates are, by definition, nearly independent of the cluster structure, hence they increase the variance within clusters. In addition, nuisance features can, by chance, create spurious similarities between points belonging to different clusters. Altogether this shortens the cluster exit time.
> 3. **Randomness**- The algorithm is indeed random, as stated in the first step of Algorithm 1. We present the standard deviation of the clustering performance on real datasets in the supplementary material (see Table S1). Furthermore, in figure S4, we present the average performance over 100 simulations. Following this comment, we will add statistical measures to analyze the randomness of the method on synthetic examples.
> 4. **Future works** - We agree. We have several future directions we are pursuing extending the proposed methodology. Specifically, since the method only focuses on low-frequency features, we design an additional selection term to choose representative features from a subset of correlated features. Furthermore, we are studying the potential applications of the method to several challenging biological tasks, such as cell cycle analysis and identification of developmental tree structures. Specifically, we recently demonstrated that the proposed scheme can identify a subset of features related to the cell cycle in single-cell RNA-Seq data consisting of thousands of cells. Embedding the data using this subset of selected features (genes) reveals a circular manifold structure of the data points (cells included in the experiment). We further verify that the identified genes coincide with known markers for the cell cycle. Importantly, our method is able to discover these genes in a completely unsupervised fashion. We will elaborate on this result in the revised version.
> **Summary** - The response above is aimed to clarify several concerns and help emphasize our novelty & merits to the reviewer. We want to thank the reviewer again for acknowledging the significance and clarity of our work and for helping us improve the paper. We hope that in light of these clarifications, the reviewer may consider increasing his score and voting for acceptance of the manuscript.

---

> > ### Author Response · Authors · 2021-08-19
> > **Discussion engagement**
> >
> > We would like to thank the reviewers again for their valuable comments. We hope that our response helped clarify all of the concerns raised by the reviewers. If the reviewers have any additional questions or points which require clarification, we would greatly appreciate any engagement in the open review discussion.

---

### Official Review · Reviewer_LQQD · 2021-07-16

**Rating:** 6
**Confidence:** 3

**Summary:**

This paper proposes a novel unsupervised feature selection method to efficiently remove nuisance features. The model combines a graph Laplacian score with stochastic gates, which adds a re-evaluation of the graph Laplacian to improve the performance in downstream applications. Experiment results are provided to validate the theoretical claims.

**Ethical Concerns:**

n.a.

**Limitations And Societal Impact:**

See Main Review.

**Main Review:**

Pros:

1. The proposed method applied differential stochastic gates to compute the Laplacian scores, which added a novel perspective to identify the informative features with some theoretical assumptions.

2. Removing nuisance features is a fundamental task when dealing with high-dimension data, which is a useful method to discover main structure and improve the performance of clustering.


Cons/Questions:

1. The presentation is somewhat difficult to follow. The writing could be improved by framing the storyline more, giving the readers more guideposts and simplifying the caption of figures.

2. It would be interesting to see the performance on the manifold learning task.

3. It would be better to try different noise patterns. For real data, entire dimensions may be more complicated including some structured but useless features. A multivariate Gaussian distribution can be considered as one of common noise patterns. More noise levels and patterns can be investigated.

4. In the experiments of real-world data, what is the general relationship between the clustering accuracy and dimension of selected features? What will the results change if all the baselines choose the same dimensions of features? As a hyper-parameter, is there any quick selection mechanism or theoretical guarantee for the dimension/proportion of the selected features?


**Time Spent Reviewing:**

~5

---

> ### Author Response · Authors · 2021-08-10
> **Response to Reviewer #2 - LQQD**
>
> We would like to thank the Reviewer for the feedback, suggestions, and constructive comments. We provide our response below:
> 1. **Presentation**- We thank the reviewer for pointing out the need for improvement in the presentation and figure captions. To address this comment, sections 1-4 will be edited to improve the framing of our proposed methodology. Furthermore, the captions of figures 1-4 are being rewritten to help the reader interpret our results.
> 2. **Feature selection for Manifold Learning** - This is a good point. Our ongoing evaluation of the method's benefits for manifold learning in biology has been successful. Specifically, we recently demonstrated that the proposed scheme can identify a subset of features related to the cell cycle in single-cell RNA-Seq data consisting of thousands of cells. Embedding the data using this subset of selected features (genes) reveals a circular manifold structure of the data points (cells included in the experiment). We further verify that the identified genes coincide with known markers for the cell cycle. Importantly, our method is able to discover these genes in a completely unsupervised fashion. We will elaborate on this result in the revised version.
> 3. **Other noise models** - We absolutely agree and are currently evaluating the method on different noise structures, including noise with a correlated covariance matrix.
> 4. **Clustering accuracy vs. the number of selected features** - There is no single relationship between performance and the number of selected features for the real datasets. In the supplemental material section, we partially addressed this question by presenting the performance for several values of selected features across three real datasets (see Figure S5). The param-free loss enables the selection of smooth features without tuning this hyperparameter. Another way to choose the number of selected features in a completely unsupervised fashion is by evaluating the average Laplacian Score (over the set of selected features) for different numbers of selected features and choosing the amount that maximizes the average Laplacian Score.
>
> **Summary** - We hope the response above clarifies several concerns and helps emphasize our novelty & merits to the reviewer. We would like to thank the reviewer again for acknowledging the significance of our work and for helping us improve the paper.

---

### Official Review · Reviewer_Rzva · 2021-07-17

**Rating:** 7
**Confidence:** 4

**Summary:**

The paper proposes a feature selection mechanism that aims to preserve the structure of the neighborhood graph of the data. The mechanism is based on the use of stochastic gates that is trained via a regularizer, which balances (i) the goodness of fit of the selected features to the neighborhood graph Laplacian with (ii) the expected value of the number of features preserved. The numerical results show improvements in performance vs. the literature of varying levels and in most tested datasets.

**Limitations And Societal Impact:**

The limitations provided are common to feature selection methods: it is assume that nuisance variables exist in the original data, and here it is further assumed that they are of a high frequency nature. The notion of frequency, however, is not described in terms of the problem addressed in the manuscript; does this refer to the alignment of the features with the eigenvectors of the graph Laplacian having small eigenvalues?

There is an interesting question here of what the dimensions that are discarded might mean in real-world datasets that are successfully modeled by neighborhood graphs. For example, while in images discarding certain pixels may not have a negative impact, one wonders what may be discarded when working with demographic data, for example.

**Main Review:**

I am curious as to how the proposed method preserves the structure of the original data given that the original data graph and the full data are not referred to in the objective functions implemented here. In [36], the features selected are used to predict the same labels as the full data. Here, it appears to me that the features are judged as to how well-matched they are to the graph structure appearing from the *currently selected* features, not the entire set available originally.

Is there a sense for how well the proposed method preserves the structure of the full data that are relevant in the unsupervised learning task? This is an interesting question in terms of how the clustering performance is preserved after the selection, as well as to  whether the proposed method can be expanded to other tasks beyond clustering.

It would be good to describe how the derivative in the equation after line 229 is computed.

The author questions state that a theoretical result is provided in Section 3.2, but no new lemma or theorem appears there. Perhaps (5) should be reformatted as such.

Some of the answers to the author questions (Q4, regarding data) are missing.

**Time Spent Reviewing:**

4 hours

---

> ### Author Response · Authors · 2021-08-10
> **Response to Reviewer #1 - Rzva**
>
> We would like to thank the Reviewer for the feedback, suggestions, and constructive comments. We provide our response below:
> 1. **Relation between cluster structure and selected features** - The reviewer raises an interesting question about the relation between the cluster structure captured by the selected features and the structure represented by the complete set of features. This is partly addressed by our empirical evaluation (see table 1), which demonstrates that the selected features often capture cluster structures that are more consistent with the actual labels. Nonetheless, we are currently pursuing an extension of the method which can select features that lead to improved manifold learning. To that end, an evaluation of the method's benefits for manifold learning in biology is currently underway. In fact, we have already observed that the proposed scheme can identify a subset of features related to the cell cycle in single-cell RNA-Seq data consisting of thousands of cells. Embedding the data based on this subset of selected features (genes) reveals a circular manifold structure of the data points (cells included in the experiment).
> 2. **Reframing the theoretical part** - Correct, we will rephrase (5) as a new proposition and add the missing details in response to Q4. Regarding Q4 (a), we cite all the authors who collected the datasets used in our paper within the supplemental material section, S7.  Questions (b)-(d) these questions are irrelevant for our datasets since no new assets are presented in the paper  (N/A). (e) the data we are using does not contain identifiable personality information or offensive content.
> 3. **Notion of high frequency** - Correct, we refer to high-frequency features as those which agree highly with eigenvectors of the graph Laplacian having small eigenvalues. This point will be clarified in the manuscript.
> 4. **Feature selection on demographic data**- This is an exciting point, especially for practitioners. For example, in the biological data BIASE, the method removes 25,383 genes and leads to perfect clustering accuracy on the four cell types in the data. This dramatically exceeds the 41.8% clustering accuracy obtained from applying K-means to the complete set of features. Such an improvement in accuracy suggests that some of the discarded genes have no relation to the true cell identity and might, consequently, have no smooth structure with respect to the data graph. Following this suggestion, we performed an experiment on the ADULT dataset*, containing demographic data. This is binary data, with a label indicating if the household income exceeds 50k$ per year. By applying DUFS to the continuous variables we observe that the “Capital-gain” is ranked highest (in terms of smoothness by DUFS) while “Final Weight” (which indicates the number of individuals with a certain set of features) is ranked lowest. The clustering accuracy in fact increases by ~2% when removing the values of the “Final Weight” from the data.
> *https://archive.ics.uci.edu/ml/datasets/adult
>
> **Summary** - We hope the response above clarifies several concerns and helps emphasize our novelty & merits to the reviewer. We would like to thank the reviewer again for acknowledging the significance of our work and for helping us improve the paper.

---

### Comment · Area_Chair_UfF2 · 2021-08-28
**Role of different Laplacians and of gate stochasticity**

I would have two clarification questions for the authors:

1.  $L_{un}$ *vs* $L_{diff}$

The Laplacian Score (LS) of a feature $f$ is defined in this paper as $f^\top L_{un} f$ where $L_{un}=D-K$ but the Laplacian used in the algorithm proposed in the paper is $L_{diff}$ which leads to the score $f^\top L_{diff} f.$ These two scores are obviously not equivalent.
Now the focus of this paper is on the introduction of *stochastic gates* and the paper aims at showing that the *stochastic gates* produce a better selection of features. However, if the scores used in the first place in the baseline and in the proposed method are not the same, I am not sure to understand how to know if the improvement is due to the change of Laplacian or to the introduction of the stochastic gates. How does the choice made by the authors affect the conclusion of the paper? Could the authors clarify this?

(As a side note the Laplacian Score introduced in the original paper, He et al. (2006), has two variants: the first is $\frac{f^\top L_{un} f}{f^\top D f}$ and the second is $f^\top L_{un} f$ if we assume that the variable have unit variance. There is no discussion in the submitted paper of why the authors choose to consider only the second form.)

2. **Stochastic gates *vs* Deterministic gates**

The probability model for the stochastic gates depends on a parameters $\sigma$ which defines how noisy the gates are (see equation (7)). My impression is that this value is *never* specified and that there is *no* study of the influence of this parameter in the paper. This value $\sigma$ however determines the level of stochasticity of the gate. If $\sigma=0$ then the value of $z_i$ is deterministic and the "gate" is just multiplying the data by the scaling factor $\mu_i$. If $\sigma$ is larger, then the effect of the gate is really stochastic and thus fluctuates stochastically from one iteration to the next. It seems that it would be useful to know if the stochasticity is actually of any use, as it potentially makes the formulation and the optimization more complicated, and most likely slows down the convergence of the algorithm. Could the authors clarify the situation on this point? If the stochasticity is not necessary, the formulation would be somwhat related to the *non-negative garrote*  (Breiman, L. (1995) Better subset regression using the nonnegative garrote. Technometrics, 37, 373–384.)

---

> ### Author Response · Authors · 2021-08-31
> **Response to Area Chair**
>
> We thank the area chair for these valuable questions and comments. Here, we will provide additional details to clarify the raised concerns.
>
> **1.1. $L_{un}$ vs. $L_{diff}$ :** First, we want to point out that in our manuscript, we prove that the LS breaks in the presence of many nuisance variables (see section 3.2). Our arguments and analysis hold both for $L_{diff}$ and $L_{un}$. For example, in Figure 2 we show that the second largest eigenvalue of $L_{diff}$ decreases as the number of nuisance features increases. In analogous, for $L_{un}$ the second smallest eigenvalue would increase when we add more nuisance features. This means that both Laplacians get corrupted in the presence of many nuisance features and hence the LS can become non-informative in this scenario regardless of the choice of Laplacian. Furthermore, our proposed solution to filter nuisance features before the calculation of the Laplacian is valid both for $L_{diff}$ and $L_{un}$.
>
> We chose $L_{diff}$ rather than $L_{un}$ because this variant of the Laplacian can be naturally combined with a regularizer and enable the selection of a sparse subset of smooth features. Specifically, since smooth features translate to a high score when using $L_{diff}$,  the tradeoff between the LS term, which pushes all gates to open (see Eq. (8)), and the sparsity term, which pushes all gates to close is what creates the training dynamics of DUFS. When replacing $L_{diff}$ to $L_{un}$ (see Eq. (8)), the model would select non-smooth features. Alternatively, using this replacement but with a positive sign before the first term would lead to the selection of no features.
>
>
> To demonstrate that the different Laplacian does not change the performance of LS substantially, we have reevaluated the clustering accuracies on all datasets based on $L_{diff}$ instead of $L_{un}$. In the following table, we report the results of our new experiments:
>
> | Dataset   | LS (UN) | LS (DIFF) | DUFS |
> |-----------|---------|-----------|------|
> | GISETTE   | 78.5    | 64.5      | 99.5 |
> | PIX10     | 76.6    | 81        | 88.4 |
> | COIL20    | 60      | 57.2      | 65.8 |
> | Yale      | 42.7    | 43        | 47.9 |
> | RCV1      | 54.9    | 54.9      | 62.2 |
> | TOX-171   | 47.5    | 47.8      | 49.1 |
> | ALLAML    | 73.2    | 72.2      | 74.5 |
> | PROSTATE  | 56.8    | 56.8      | 64.7 |
> | SRBCT     | 41.1    | 41.6      | 51.7 |
> | BIASE     | 83.8    | 83.7      | 100  |
> | INTESTINE | 43.2    | 43.3      | 71.9 |
> | FAN       | 42.9    | 41.7      | 49   |
> | POLLEN    | 46.9    | 46.8      | 60.2 |
>
> As evident from this table, the performance of the LS does not change substantially after changing the Laplacian from $L_{un}$ to $L_{diff}$ (except from the first two examples). This is also validated by the fact that there is an overlap of between 90%-100% (across all datasets) in the selected features when the model selects features based on $L_{un}$ or $L_{diff}$. Our model (DUFS) leads to a substantial improvement in clustering accuracies compared to both of these variants of the LS. Therefore, we argue that this improvement could not be achieved without our proposed features sparsification technique.
>  **1.2.  Different normalizations of the LS** Indeed in our methodology, we decided to focus on the variant of the LS with $f^\top L_{un} f$ computed based on normalized features (not weighted by the degree matrix D). This is because we calculate the score in each epoch. Therefore, to reduce computations we exclude the additional weighted normalization term (in $\frac{f^\top L_{un} f}{f^\top D f}$) and normalize the data once in advance. This normalization does not play a crucial role in the performance of the method. We will discuss and clarify this issue in the revised version.
>
>
>
>
>
>
>
>
>
> **2. Stochastic gates vs. Deterministic gates**: To understand the importance of stochasticity in our proposed procedure, let’s first consider a simple deterministic alternative. We could have replaced our gates with deterministic parameters (scalars that scale the features) and encouraged sparsity using an $\ell_1$ based regularization (similar to LASSO). However, such a procedure would not lead to a binary selection of variables. Instead, the $\ell_1$ would lead to shrinkage of the scaling coefficients. Using binary coefficients with an $\ell_0$ norm regularizer could overcome this limitation and lead to a true sparsification of features. However, the $\ell_0$ norm is not differentiable. Fortunately, several recent works have demonstrated that using a continuous approximation of binary probabilistic variables is effective for model sparsification [1-3] and works better than the LASSO for supervised feature selection (see [4]). Intuitively, using binary probabilistic variables can help explore the space of sparse solutions in a differentiable fashion.
>
> For this reason, we chose to use a continuous approximation of Bernoulli variables (which were demonstrated effective for supervised feature selection [4]). The injected noise (with sigma>0) is required to evaluate the LS based on different subsets of randomly chosen features. This is because the injected noise allows sampling values of Z’s (gates) that sparsify the feature set before the model converges.  Another added value of stochasticity is that it allows the model to re-evaluate a feature even if it is dropped out at an early stage of training. For these reasons, we need $\sigma>0$. Our manuscript followed the justification in [4], which argues that $\sigma=0.5$ would maximize the gradient size at initialization. Nonetheless, as stated by the AC, larger values could also work but would slow down convergence. We will discuss and clarify this point in the revised version.
>
> To provide additional intuition on the role of the injected noise ($\sigma$), we would like to refer the AC to an ongoing line of research on the role of injected noise in training neural networks [4-7]. Based on numerous experimental results, we believe that the injected noise smoothes the landscape of our loss function. This was observed by evaluating performance using smaller values of $\sigma$. This is also evident in the supervised setting studied in Figure 4 of [4]. The following paper sheds some light on this phenomenon [5]. We concur with the AC that this is an exciting question, tightly related to the success of stochastic gradient descent [6-7], and to popular tricks for training binary neural networks [8]. We will try to address this question in future research.
>
> [1] Louizos et al. Learning sparse neural networks through $ L_0 $ regularization.
>
> [2] Jang et al. Categorical reparameterization with gumbel-softmax.
>
> [3] Abid et al. Concrete Autoencoders for Differentiable Feature Selection and Reconstruction.
>
> [4] Yamada et al. Feature Selection Using Stochastic Gates.
> [5]  Zhou et al. Towards Understanding the Importance of Noise in Training Neural Networks.
>
> [6] jin et al. On the local minima of the empirical risk.
>
> [7] Kleinberg et al. An alternative view: When does sgd escape local minima?
>
> [8] Hubara et al. Binarized Neural Networks.

---

> > ### Comment · Area_Chair_UfF2 · 2021-09-12
> > **Convinced for the choice of Laplacian not concerning stochasticity**
> >
> > Dear authors,
> >
> > Thank you very much for your response, and for the effort you put in backing your arguments with numerical values, and appropriate references.
> > 1. I am entirely convinced by your argument concerning $L_{un}$ *vs* $L_{diff}$, that are supported by compelling experimental values.
> > 2. Concerning the question of stochasticity, the set of references that are provided by the authors together with their explanations indeed support well their approach. So thanks for these as well. I had in mind precisely the type of concave penalization which [4] calls DNC for *deterministic non-convex* and [4] provides a comparison is in favor of the stochastic gates... Adding a comment on this point in the paper could be useful. I am however still tempted to think that deterministic penalties like a form of binary entropy over a continuous variable $z_i \in [0,1]$ might also work well in the considered context, because it would also clip variables to $0$ or $1$.

---

> > > ### Author Response · Authors · 2021-09-19
> > > **Reply to AC**
> > >
> > >  Since the discussion period is over, below, we provide a brief response to the AC.
> > >
> > > We truly appreciate the time the AC invested in reading our response and the related material. Thanks for the valuable comments; we will add a discussion on the role that stochasticity plays in the revised version of the manuscript. We agree with the intuition of the AC that other deterministic penalties may also be beneficial for the problem of unsupervised feature selection. We believe that our work motivates sparsifying the Laplacian before the computation of the Laplacian Score and that stochasticity serves as one effective solution. At the same time, other solutions might also be beneficial. We will discuss this in the revised version.

---

### Decision · Program_Chairs · 2021-09-27

**Decision:**

Accept (Poster)

**Comment:**

The reviewers found that the proposed work proposes a sufficiently novel and interesting analysis and approach to solve an important problem, together with compelling experiments.

The rebuttal provides clear responses to the comments and questions of the reviewers.

The authors are strongly encouraged to take into account the comments of the reviewers and the elements they themselves contributed to this discussion when preparing the final version of the manuscript.